# Is There Any Difference in the Quality of CPR Depending on the Physical Fitness of Firefighters?

**DOI:** 10.3390/ijerph20042917

**Published:** 2023-02-07

**Authors:** HyeonJi Lee, JiWon Ahn, Youngsoon Choi

**Affiliations:** 1Department of Emergency Medical Technology, Kangwon National University, 346 Hwangjo-Gil, Samcheck-si 25949, Republic of Korea; 2Department of Emergency Medical Rehabilitation, Kangwon National University, 346 Hwangjo-Gil, Samcheck-si 25949, Republic of Korea; 3Department of Nursing, Kangwon National University, 346 Hwangjo-Gil, Samcheck-si 25949, Republic of Korea

**Keywords:** firefighters, physical fitness, CPR, quality of CPR, simulator

## Abstract

(1) Background: The purposes of this study were to develop a physical fitness evaluation program for new firefighters, to investigate whether there is a quality difference in performing CPR for cardiac arrest patients according to physical strength, and to provide basic data to improve CPR quality. (2) Methods: The subjects of this study were fire trainees who were appointed as firefighters for the first time in G province from 3 March 2021 to 25 June 2021. The age of the subjects was 25–29 years old, and their experience of working as a firefighter was less than three months. According to the purposes of the study, the researcher composed the Physical Fitness Evaluation Program, including the physical fitness evaluation method and steps, and requested a content expert group to modify and supplement the ‘physical fitness assessment program’. The subjects were divided into four groups according to their levels of physical strength, and CPR was performed for 50 min in groups of two. A high-end Resuscitation Anne Simulator (Laeadal, Norway) mannequin was used to evaluate the quality of CPR. (3) Results: When comparing the difference in CPR quality, there were statistically significant differences in the number of chest compressions and compression depth, but all groups met the CPR guidelines. In the case of this study, it is thought that high-quality CPR could be performed because the subjects’ average age was low and they continued to exercise to improve their physical strength for their role. (4) Conclusions: It was concluded that the fitness level of new firefighters confirmed by this study was sufficient for general high-quality CPR. In addition, for high-quality CPR, continuous management is required by developing a continuous CPR education and physical training program for all firefighters.

## 1. Introduction

The survival rate of cardiac arrest in Korea is increasing every year. Since the first acute cardiac arrest investigation began in 2006, the survival rate improved from only 2.3% in 2006 to 8.7% in 2019, and the recovery rate of brain function improved from 0.6% to 5.4% [1]. However, these rates are still lower than those in advanced countries’ emergency medical centers, and the improvement has slowed somewhat in recent years. This is the result of the gradual development of the medical field’s cardiac arrest research, social awareness and system development, and the firefighting service centered on the 119 paramedics.

In the past, cardiopulmonary resuscitation (CPR) education involved teaching basic CPR at the level of the public. As the social necessity of high-quality CPR according to the changes of the times has emerged, a change in the educational system has been demanded [2,3].

Firefighters exposed to pre-hospital emergencies should apply standardized CPR training to firefighters and rescuers and firefighters with professional qualifications, and train them to overcome their physical limitations [4,5]. In order to develop an educational curriculum with an educational goal of dealing with this changing external environment, a series of tasks to evaluate, analyze, and supplement each detailed aspect of CPR are needed. However, research on the ability to sustain CPR according to the educational standards and physical strength for pre-hospital CPR, especially on-site CPR for firefighters, is insufficient.

Mickey Eisenberg of the University of Washington Resuscitation Academy in Seattle, Washington, a region with one of the highest cardiac-arrest survival rates in the world, suggested the Utstein ten-steps implementation strategy (UTIS) as a specific method to improve the cardiac-arrest survival rate in the region. Among the ten steps, the items that best matched the domestic field ambulances in steps one to four, which consist of four steps for basic life support (BLS), were high-performance CPR and rapid dispatch. A previous study analyzed the effect on the survival rate of applying the UTIS CPR package to the field from 2015 to 2016 in one region in Korea [6]. Positive effects were found for many indicators, and prehospital circulation recovery, survival rate, and good cerebral performance category (CPC) ratio improved. In particular, the recognition time for cardiac arrest in the control room was greatly reduced, and the frequency of use of advanced cardiovascular life support (ACLS) and specialized techniques has increased greatly due to the multiple dispatch system [6]. Due to the nature of the dispatches of firefighters, pre-hospital CPR must be performed in a relatively poor environment for a long time and with few personnel [7].

It is very important for firefighters to improve and maintain physical strength. Due to the nature of firefighters’ jobs, they have to move while wearing heavy and bulky protective clothing and breathing apparatus in dangerous places, they have to use great force when rescuing patients, and the working environment is often not guaranteed [8,9]. Firefighters must maintain an appropriate level of fitness to meet the high-intensity physical demands of their job. However, there are cases where rescue and treatment are performed without proper physical conditions due to lack of time for exercise to improve physical strength, and in this case, the ability to perform work may deteriorate [10,11]. According to some studies, firefighters require a higher level of physical strength than the public, and there are studies showing that a high level of physical strength has a positive effect on job performance [12,13].

For firefighters, muscular strength and muscular endurance are important elements of physical fitness. In the past, firefighters’ CPR training involved basic CPR at the general level. As the social need for high-quality CPR has emerged, changes in the educational system have been required [2,3].

Standardized CPR training should be applied to all firefighters (paramedics with professional qualifications and firefighters and rescuers) who are exposed to emergencies before hospitals, and they should be trained to overcome physical limitations [5,7]. In order to develop the educational curriculum to meet these changing external environments, a series of tasks to evaluate, analyze, and supplement each detailed aspect of CPR is required. However, there are very few studies on the CPR continuity competency according to educational standards and physical strength for pre-hospital CPR, especially on-site CPR for firefighters. Although one needs to pass the physical fitness test to become a firefighter, there is no standardized education program to improve physical fitness after becoming a firefighter, and there is no exercise program tailored to each individual’s physical requirements.

We aimed to develop a physical fitness evaluation program for new firefighters, compare and analyze how firefighters’ physical fitness levels affect the quality of CPR, and provide basic data for future CPR training and to improve the survival rate of out-of-hospital cardiac arrest (OHCA).

## 2. Materials and Methods

### 2.1. Research Subjects and Design

The subjects of this study were approached between 3 March 2021 and 25 June 2021. They were firefighter trainees who were first appointed as firefighters in G province. The age of the subjects was 25–29 years old, and their experience in the job was less than three months. Subjects who had received CPR training within 6 months were excluded. The number of samples required for one-way ANOVA was calculated using the G*Power 3.1.9.4 program. When the significance level was 0.05, the power was 0.95, the effect size was 0.5, and there were 4 groups; the calculated sample size was 10 subjects per group. In this study, 20 subjects per group were selected in anticipation of dropout.

### 2.2. Physical Fitness Assessment Program

To compose the ‘physical fitness assessment program’, a program content expert group was composed of one emergency rescue professor, one nursing professor, and two first aid technicians with more than 10 years of experience. According to the purposes of this study, the researcher composed the physical fitness evaluation program, including the physical fitness evaluation method and steps, and the content expert revised and supplemented the written fitness evaluation program. See Table 1.

The Physical Fitness Evaluation Program evaluated 185 new firefighter trainees. The criteria for the physical fitness evaluation grade according to the physical fitness measurement result are in Table 2. One-hundred and eighty-five new firefighters were classified into four groups (Group A: best, Group B: high, Group C: medium, Group D: low) according to their grades in the Physical Fitness Evaluation Program. After that, 20 p were randomly assigned to each group. The classification of the four groups was established by the content expert group of the ‘Physical Fitness Evaluation Program’.

The trainees’ CPR scenario was conducted based on the ‘Basic CPR algorithm among the 2020 guidelines’ of the Korean CPR Association. The brief order is ‘Check patient response → Report → Check respiration and pulse → Chest compression and artificial respiration → After attaching the defibrillator, alternately perform CPR according to the instructions of the defibrillator at two minute intervals’.

The evaluation time of CPR quality was divided into 0–10 min/10–20 min/20–30 min/30–40 min/40–50 min. For each interval, the average chest compression depth, compression rate, compression rate, and relaxation rate, and the appropriate chest compression position, were numerically recorded.

### 2.3. Research Tools

A high-end Resuscitation Anne Simulator (Laeadal, Stavanger, Norway) mannequin was used to evaluate the quality of CPR in this study. With SIMPAD, the following detailed indicators can be immediately fed back through the screen, and data can be utilized and managed because of the storage function. In this study, the following CPR quality indicators were measured (see Table 3).

### 2.4. Data Analysis Method

For analysis of the collected data, SPSS window 23.0 was used. One-way ANOVA was used to compare the quality of CPR according to the physical fitness group. In this study, it was determined that there was a statistically significant difference when the *p* value was less than 0.05.

### 2.5. Ethical Considerations

Respondents who understood the purpose of this study and gave written consent to participate were selected. In order to consider the ethical aspects of the respondents, the purpose of this study and data collection methods were explained, and the subjects were those who voluntarily participated in data collection and gave written consent to participate in the study. They were informed that they could withdraw from participating in the study at any time while the study was in progress. The ethical aspects of the respondents were considered by explaining that the collected data would be anonymous and only used for research purposes, and that there would be no exposure of personal information and no disadvantages.

## 3. Results

### 3.1. Comparison of CPR Quality by Physical Fitness Group (0–10 min)

There was no statistically significant difference when comparing the CPR quality by physical fitness group within 10 min (see Table 4). Since there was no statistically significant difference in CPR quality between 11 and 20 min among fitness groups, the table was not inserted.

### 3.2. Comparison of CPR Quality According to Physical Fitness Group (21 to 30 min)

For the comparison of CPR quality according to the physical fitness group, the comparison results for between 21 and 30 min were as follows (Table 5). There was no statistically significant difference in the quality of CPR among physical fitness groups between 21 and 30 min.

### 3.3. Comparison of CPR Quality by Physical Fitness Group (31 to 40 min)

In the comparison of CPR quality according to the physical fitness group, the comparison results for 31 to 40 min were as follows (Table 6). There were statistically significant differences in the number of compressions per minute and compression position between 31 and 40 min, between physical fitness groups. The number of compressions per minute according to physical strength: Group A = 112.60 ± 7.486. Group B = 121.60 ± 5.125. Group C = 117.40 ± 8.449. Group D = 117.43 ± 4.995. Group B showed statistically significantly higher scores (F = 2.913, *p* < 0.05). Compression position (%) according to physical strength: Group A = 76.80 ± 22.080. Group B = 90.30 ± 12.473. Group C = 68.80 ± 25.806. Group D = 58.00 ± 29.235. There were statistically significant differences (F = 3.108, *p* < 0.05). The number of compressions per minute was significantly lower in Group A than in Groups B, C, and D, and in particular, Group B participants did 121.60 on average, which was out of the standard range. The compression position was the most accurate for Group B at 90.3%.

### 3.4. Comparison of CPR Quality according to Physical Fitness Group (41~50 min)

For the comparison of CPR quality according to the physical fitness group, the comparison results for between 41 and 50 min are as follows; see Table 7. The number of compressions per minute according to physical strength: Group A = 111.50 ± 6.719. Group B = 121.22 ± 4.631. Group C = 118.60 ± 7.633. Group D = 119.50 ± 3.251. Group B had a high score, which is statistically significantly different from the others (F = 4.304, *p* < 0.05). Compression position (%) according to physical strength: Group A = 83.50 ± 28.894, Group B = 86.89 ± 17.603. Group C = 48.00 ± 37.824. Group D = 67.63 ± 29.554. There were statistically significant differences (F = 3.366, *p* < 0.05). Ventilation adequacy (%) according to physical fitness showed statistically significant differences: Group A = 22.17 ± 22.534, Group B = 29.33 ± 17.734. Group C = 58.50 ± 28.117. Group D = 32.00 ± 22.307’ (F = 3.625, *p* < 0.05). The number of compressions per minute was significantly lower in Group A than in Groups B, C, and D, and in particular, Group B participants performed 121.22 on average, which was out of the standard range. The compression position was the most accurate in Group B at 86.89%.

### 3.5. Comparison of CPR Quality for Total Time according to Physical Fitness Group

The comparison results for the whole period are as follows; see Table 8. The average compression depth according to physical strength: Group A = 54.83 ± 4.635, Group B = 56.66 ± 3.472, Group C = 57.10 ± 4.483, Group D = 57.00 ± 2.762. The differences were statistically significant (F = 3.387, *p* < 0.05). The number of compressions per minute according to physical strength: Group A = 111.89 ± 7.889, Group B = 119.28 ± 6.310, Group C = 116.37 ± 8.99, Group D = 115.66 ± 6.394. Group B had a high score which is statistically significantly different from the others (F = 7.599, *p* < 0.001). The speed (%) according to physical fitness: Group A = 70.07 ± 27.493, Group B = 50.17 ± 32.206, Group C = 62.02 ± 31.933, Group D = 63.68 ± 23.986. The differences were statistically significant (F = 3.759, *p* < 0.05). Compression position (%) according to physical strength: Group A = 72.28 ± 24.865, Group B = 86.19 ± 17.975, Group C = 62.78 ± 31.591, Group D = 64.11 ± 26.631. The differences were statistically significant. (F = 8.166, *p* < 0.001). Insufficient ventilation (%) according to physical strength had statistically significant differences: Group A = 51.41 ± 32.295; Group B = 36.56 ± 25.694; Group C = 43.47 ± 27.033; Group D = 56.11 ± 35.434 (F = 3.070, *p* < 0.05). Ventilation adequacy (%) according to physical fitness had statistically significant differences: Group A = 17.56 ± 18.433, Group B = 25.26 ± 19.025, Group C = 46.11 ± 28.327, Group D = 29.33 ± 24.326 (F = 10.082, *p* < 0.001). Ventilation excess (%) according to physical fitness had statistically significant differences: Group A = 40.03 ± 29.691, Group B = 40.24 ± 29.554, Group C = 17.16 ± 17.700, Group D = 49.05 ± 37.815. Note the differences (F = 5.231, *p* < 0.05). The chest compression depth (54.83 mm) and the number of compressions per minute (111.89) were lower in Group A than the other groups, but all groups met the guidelines. The exact compression position for Group B (86.19%) had the highest value. The ventilation volume was recorded for Groups A and B, which was an appropriate ventilation volume of 500–600 cc.

## 4. Discussion

This study targeted new firefighters in G province. They were divided into Groups A, B, C, and D in ascending order according to their physical fitness level, and the quality of CPR was measured over a 50-min period and compared in 10 min increments.

There was no significant difference between groups until 30 min into CPR. From 31 to 50 min, the number of compressions per minute in Group A was maintained at an appropriately constant rate, more so than in Groups B, C, and D. However, in Groups C and D, the chest compression rate of CPR was 100–120 times per minute, which is within the normal range, and the average rate was 121.60 in Group B. Although Group B was on the good side of physical fitness, performance is thought to be related to the individual’s ability to perform CPR rather than being a problem of physical strength. However, the accuracy of chest compression position was the most accurate in Group B at 90.3%.

Overall, there were statistically significant differences in the number of compressions per minute among groups, but the compression depth and number of compressions per minute in all groups were in accordance with the CPR guidelines, and other indicators had no significant differences. CPR guidelines recommend alternating for chest compressions every 2 min. In previous studies, it was reported that the depth of chest compression significantly decreased between 90 s and two minutes due to physical strength [14,15], and another study reported that compression adequacy decreased by 18.6% per minute after the first minute of compression [16]. No such change was observed in this study. Rather, it showed a similar pattern to that of previous studies in the CPR ability of second-class emergency responders, in which no change in the chest compression success rate was observed during four cycles [17]. The subjects of this study were new firefighters, most of them in their 20s and 30s, and it is considered that their physical strength is better than that of the public. There were changes during CPR for 50 min, but there was no significant deviation from the guidelines or a sharp decrease in CPR quality compared to the initial stage. In addition, the continuous feedback for communication between team members and standardized two-person CPR seems to have been helpful. This has also been described in other studies on the effect of team-based CPR [4,18,19]. Through these results, it was found that there was no significant difference based on physical strength in maintaining high-quality CPR. In this regard, we think that the same basic CPR guidelines should be applied to CPR from the public to experts.

Pre-hospital cardiac arrest happens in many unpredictable and difficult situations, and like all on-site firefighters, paramedics often require extreme physical strength. CPR in a mountain rescue or hardship rescue situation must be different from the CPR standards of hospitals, and all firefighters who perform it must have appropriate physical strength, and continuous management of CPR training is required. Among the previous studies related to the physical fitness of firefighters, there is a study showing that grip strength and back strength can increase the depth of chest compression in CPR [20]. In addition, studies on the correlation between physical fitness and CPR are continuously emerging, such as one study [21] showing that the body weight of a rescuer and upper body muscle mass each have an effect on chest compression. As such, there is a close relationship between physical strength and CPR for firefighters who are frequently exposed to dangerous environments and emergencies compared to the public, and it is necessary to systematically study the physical strength training and management methods that affect CPR for firefighters in the future.

In addition, until recently, the practice mannequin that has been widely used in CPR education is a low-performance mannequin. Therefore, the evaluation of chest compression and ventilation depended on the subjective judgment and sensations of the instructor, and there is a limit to the evaluation of CPR and accurate and consistent feedback. Although it is difficult to conduct training in various scenarios, the recently introduced high-performance mannequin is developing into a form that can accurately record the parameters and numbers of chest compressions and ventilations related for standardized CPR [22]. Previous studies recommend using a device that records values such as chest compression depth and speed and provides immediate feedback [23,24]. It is also mentioned that the use of high-fidelity manikins during CPR can increase the effectiveness of education [25,26]. Therefore, education using high-fidelity manikins should be expanded for various specialized scenarios.

Separate physical training is also important to improve CPR skills, and it is necessary to develop and distribute customized physical fitness management programs for field paramedics, such as stretching and improvement methods to relieve shoulder and wrist pain before and after CPR training. In addition, a continuing education program on CPR for firefighters should be applied. Researchers should acquire and apply up-to-date CPR knowledge and skills to contribute to increasing the pre-hospital resuscitation rate.

## 5. Conclusions

The purposes of this study were to develop a physical fitness evaluation program for new firefighters, to investigate whether there is a quality difference in performing CPR on cardiac arrest patients according to physical strength, and to provide basic data to improve physical strength and CPR quality.

The quality of chest compressions according to physical strength was different in some groups, but overall, it did not deviate from the standard range, despite moderate physical exhaustion. This is because the experimental group was composed of new firefighters, so the overall physical fitness level was very high. In addition, although additional follow-up studies targeting front-line firefighters are needed, it was concluded that the fitness level of new firefighters confirmed by this study was sufficient for general high-quality CPR. Due to the nature of firefighters who are exposed to various situations and unstable environments before hospitals, physical fitness factors are very important. However, even if there is a lack of physical fitness, standardized, high-quality CPR is possible for anyone, and it is thought that it is possible to overcome physical limitations with will and mental strength. In this study, it was also helpful to maintain high-quality CPR through a mannequin with a feedback device attached. In addition, it is necessary to evaluate and analyze the current quality of trainees using specialized training equipment with an educational data collection function. Based on the collected data, direct education to improve the quality of CPR of trainees can be provided.

## Figures and Tables

**Table 1 ijerph-20-02917-t001:** Physical fitness program.

Procedure	Method
1	From the starting line, walk down the road for 2 km
2	Return to the starting point, put on a respirator set (removal of face mask and auxiliary mask, air filling pressure of 200 bar or more), and walk 2 km on the road again
3	Afterwards, carry a heavy object (16 kg kettlebell, 2 pcs) placed at the designated location and travel 75 m round-trip.
4	Hold one 65 mm water tube in each hand (no restriction on carrying posture). Steps from the parking lot next to the playground → Stairs to the main building → Stairs to the training ground → Measure the time until your feet pass through the last steps of the auditorium. (total number of steps is 100).

**Table 2 ijerph-20-02917-t002:** Classification table of physical fitness by group.

Grade	Classification	Physical Fitness Record	Number of People Assigned
A	best	20~22 min 59 s	20
B	high	23~24 min 59 s	20
C	medium	25~26 min 59 s	20
D	low	27~30 min 48 s	20

**Table 3 ijerph-20-02917-t003:** Detailed measurement indicators of mannequins for professional practice.

Indicators	Indicators
▶ overall CPR Comprehensive Score (%)	▶ chest compression position accuracy (%)
▶ ventilation inflow accuracy (%)	▶ sufficient/insufficient relaxation accuracy (%)
▶ average volume of ventilation (mL)	▶ chest compression depth accuracy (%) and average depth (mm)
▶ number of artificial respirations per minute (times)	▶ total number of chest compressions and number of compressions per minute (times)
▶ total number of artificial respirations (times)	▶ chest compression speed accuracy (%)
▶ total practice time and average compression stop time (seconds)	

**Table 4 ijerph-20-02917-t004:** Comparison of differences in CPR quality by physical fitness group (0 to 10 min).

Variable	Group A		Group B		Group C		Group D		F	*p*
	MD ± SD		MD ± SD		MD ± SD		MD ± SD			
average compression depth (mm)	54.90	±3.957	57.13	±2.588	56.60	±4.477	57.40	±2.675	0.967	0.420
sufficient relaxation rate (%)	47.70	±24.846	57.63	±29.428	54.80	±32.530	59.80	±23.11	0.357	0.784
sufficient compression rate (%)	59.80	±12.603	43.38	±26.742	37.70	±25.082	41.30	±24.891	1.837	0.159
number of compressions (/min)	108.70	±9.707	111.13	±4.390	112.00	±10.781	110.90	±6.173	0.280	0.839
speed (%)	59.20	±23.484	75.75	±23.879	61.70	±35.359	71.90	±18.19	0.850	0.477
compression position (%)	68.30	±22.691	76.13	±28.140	61.00	±26.373	62.40	±26.709	0.617	0.609
average ventilation volume (mL)	503.43	±188.05	504.80	±91.221	457.80	±53.616	428.44	±163.27	0.470	0.706
ventilation lack (%)	62.57	±42.18	47.60	±43.230	53.60	±23.692	67.89	±38.699	0.358	0.784
ventilation adequate (%)	3.50	±5.505	22.40	±22.579	31.20	±23.253	22.60	±29.880	1.650	0.215
ventilation excess (%)	40.17	±40.667	30.00	±29.223	15.20	±17.254	58.67	±40.067	1.212	0.339

Group A: best, Group B: high, Group C: medium, Group D: low.

**Table 5 ijerph-20-02917-t005:** Comparison of CPR quality according to physical fitness group (21 to 30 min).

Variable	Group A		Group B		Group C		Group D		F	*p*
	MD ± SD		MD ± SD		MD ± SD		MD ± SD			
average compression depth (mm)	55.89	±4.755	56.40	±3.596	58.11	±3.516	56.67	±2.500	0.607	0.615
sufficient relaxation rate (%)	50.78	±22.643	52.50	±23.628	56.00	±23.103	59.67	±9.220	0.331	0.803
sufficient compression rate (%)	30.67	±19.660	41.80	±31.996	34.00	±23.701	40.89	±14.828	0.479	0.699
number of compressions (/min)	113.11	±8.115	121.20	±5.996	117.00	±8.588	116.22	±6.591	1.956	0.140
speed (%)	68.44	±33.050	43.20	±35.627	66.56	±28.014	62.56	±22.495	1.410	0.257
compression position (%)	70.00	±24.663	86.60	±14.721	62.33	±35.313	65.33	±29.517	1.573	0.214
average ventilation volume (mL)	551.56	±143.854	560.11	±129.78	484.29	±79.573	492.88	±200.78	0.581	0.632
ventilation lack (%)	46.11	±32.289	31.00	±24.627	39.29	±26.132	57.25	±40.156	1.049	0.386
ventilation adequate (%)	23.00	±21.712	27.22	±18.593	45.57	±26.919	33.00	±23.065	1.422	0.260
ventilation excess (%)	43.00	±31.358	47.00	±32.302	26.50	±17.711	44.25	±46.850	0.363	0.781

Group A: best, Group B: high, Group C: medium, Group D: low.

**Table 6 ijerph-20-02917-t006:** Comparison of CPR quality by physical fitness group (31 to 40 min).

Variable	Group A		Group B		Group C		Group D		F	*p*
	MD ± SD		MD ± SD		MD ± SD		MD ± SD			
average compression depth (mm)	54.20	±5.350	55.90	±3.900	57.40	±3.718	56.71	±3.450	1.037	0.389
sufficient relaxation rate (%)	53.40	±23.052	57.70	±23.405	59.80	±26.191	50.14	±25.996	0.262	0.851
sufficient compression rate (%)	34.70	±16.533	41.40	±27.637	39.00	±23.007	36.57	±16.379	0.174	0.913
number of compressions (/min)	112.60	±7.486	121.60	±5.125	117.40	±8.449	117.43	±4.995	2.913	0.049 *
speed (%)	73.90	±29.118	42.20	±32.526	61.20	±32.540	63.57	±30.116	1.782	0.170
compression position (%)	76.80	±22.080	90.30	±12.473	68.80	±25.806	58.00	±29.235	3.108	0.040 *
average ventilation volume (mL)	535.38	±190.04	571.11	±124.93	492.67	±84.229	536.14	±208.54	0.387	0.763
ventilation lack (%)	45.75	±33.915	33.67	±29.866	38.00	±29.833	42.00	±32.919	0.229	0.875
ventilation adequate (%)	24.43	±18.778	24.22	±24.458	49.33	±29.614	39.67	±29.042	1.886	0.156
ventilation excess (%)	43.83	±33.517	42.11	±37.508	19.00	±26.038	42.00	±40.947	0.708	0.558

* *p* < 0.05, Group A: best, Group B: high, Group C: medium, Group D: low.

**Table 7 ijerph-20-02917-t007:** Comparison of CPR quality according to physical fitness group (41~50 min).

Variable	Group A		Group B		Group C		Group D		F	*p*
	MD ± SD		MD ± SD		MD ± SD		MD ± SD			
average compression depth (mm)	54.63	±5.502	56.11	±4.595	56.20	±5.903	56.50	±2.976	0.235	0.871
sufficient relaxation rate (%)	51.63	±29.785	60.89	±23.762	59.10	±15.808	60.88	±17.233	0.327	0.806
sufficient compression rate (%)	40.25	±30.761	34.22	±18.787	29.90	±20.174	47.75	±26.585	0.901	0.452
number of compressions (/min)	111.50	±6.719	121.22	±4.631	118.60	±7.633	119.50	±3.251	4.304	0.012 *
speed (%)	80.25	±28.009	45.56	±30.212	57.30	±36.157	55.00	±28.213	1.851	0.159
compression position (%)	83.50	±28.894	86.89	±17.603	48.00	±37.824	67.63	±29.554	3.366	0.031 *
average ventilation volume (mL)	421.33	±207.53	542.00	±101.88	499.88	±59.104	523.71	±202.56	0.865	0.472
ventilation lack (%)	50.83	±30.564	35.89	±17.989	34.57	±20.799	46.86	±32.493	0.691	0.566
ventilation adequate (%)	22.17	±22.534	29.33	±17.734	58.50	±28.117	32.00	±22.307	3.625	0.027 *
ventilation excess (%)	32.40	±22.233	39.13	±28.608	18.00	±15.232	45.00	±41.093	0.870	0.475

* *p* < 0.05, Group A: best, Group B: high, Group C: medium, Group D: low.

**Table 8 ijerph-20-02917-t008:** Comparison of CPR quality for total time according to physical fitness group.

Variable	Group A		Group B		Group C		Group D		F	*p*
	MD ± SD		MD ± SD		MD ± SD		MD ± SD			
average compression depth (mm)	54.83	±4.635	56.66	±3.472	57.10	±4.483	57.00	±2.762	3.387	0.019
sufficient relaxation rate (%)	51.76	±23.238	56.64	±25.493	57.39	±23.855	59.39	±18.282	0.905	0.440
sufficient compression rate (%)	42.85	±22.157	41.00	±26.980	34.35	±21.065	44.77	±23.258	1.770	0.154
number of compressions (/min)	111.89	±7.889	119.28	±6.310	116.37	±8.999	115.66	±6.394	7.599	0.000 **
speed (%)	70.07	±27.493	50.17	±32.206	62.02	±31.933	63.68	±23.986	3.759	0.012 *
compression position (%)	72.28	±24.865	86.19	±17.975	62.78	±31.591	64.11	±26.631	8.166	0.000 **
average ventilation volume (mL)	511.38	±164.50	551.28	±107.114	480.32	±70.809	488.08	±194.48	1.911	0.130
ventilation lack (%)	51.41	±32.295	36.56	±25.694	43.47	±27.033	56.11	±35.434	3.070	0.030 *
ventilation adequate (%)	17.56	±18.433	25.26	±19.025	46.11	±28.327	29.33	±24.326	10.082	0.000 **
ventilation excess (%)	40.03	±29.691	40.24	±29.554	17.16	±17.700	49.05	±37.815	5.231	0.002 *

* *p* < 0.05, ** *p* < 0.001, Group A: best, Group B: high, Group C: medium, Group D: low.

## Data Availability

The data presented in this study are available on request from the corresponding author.

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
