# Peer review of "Is There Any Difference in the Quality of CPR Depending on the Physical Fitness of Firefighters?"

_ijerph, 2023, doi:10.3390/ijerph20042917_

Round 1
Reviewer 1 Report
This paper has the potential to be interesting and informational however in the current form is not suitable for publication.
Abstract: overall poorly written, the background section should not summarize the study it should provide the reader with evidence as to why the study is important. Participant mean and SD need to be reported for age, height, weight, etc. There also needs to be a reference to how fitness was determined in the abstract.
The introduction fails to make any mention of physical fitness and it's importance among first responders. Overall introduction needs to be improved and make relevant and meaning connections to previous literature that discusses the importance of physical fitness in firefighters/ first responders.
Methods sections need much more detail, why is there do table with demographic information of the participants? Age? Height? Weight? How Were the fitness tests picked? Are the reliable/ sound measures? How was the classification of each fitness group determined? Are these validated/reliable classifications? What is the reference for these tests/ fitness classifications?
Results: Why is there no data for minutes 10-21? That is a large chunk of data missing and this needs to be reported. In line 176 you state that there are differences in quality of CPR by physical strength but you did not analyze just physical strength? What measure did you use? Are you just using strength or the total fitness? Group A had the lowest # of compressions but is the fittest group? This does not make sense.
Overall, the paper fails to show the importance of physical fitness in firefighters and it's connection with CPR. There is not mention of previous fitness research in the conclusion (similar to the intro). The fitness tests are poorly described and do not appear to be validated measures of fitness. Why were more standardized measures used to assess fitness? How were the fitness categories created and why? The authors need to make major revisions for this paper to be suitable for publication, in it's current state this paper is poorly written and does not provide enough information for readers to determine if scientifically sound practices/ standards were used.
Author Response
We appreciate the reviewer's valuable comments. We faithfully made revisions based on the reviewer's suggestions and comments.

Reviewer 2 Report
There is little reference to physical strength (lack of or how it may influence performance of CPR) of firefighters in the intro yet this your main variable of study. Possibly consider removing less valuable info (not that it is not important) but rather it does not lead the reader clearly to your purpose (survival rates, types of mannequins, 10 steps of UTIS). Rather I believe line 87-88 is your reasoning for exploring strength and should be elaborated upon. Please define poor environment and how that relates to strength component.
Purpose statement is confusing as written - possibly language translation?
Define quality - is this determined by depth of compressions and ability to maintain for specific period of time? This is not clear
Materials/Methods
line 99-100 - not clear as written
80 subjects total or 185 new firefighters? Unclear who the subjects actually were that were included- this info needs to be presented clearer.
line 119-120- state earlier that the group placements was determined by time to completion of physical fitness program. In addition support how this program was best determinant of physical strength necessary to perform CPR successfully.
Was 2 person CPR tested? Unclear
Ethical considerations
line148 - change to subjects- remain consistent with wording throughout
line 209-212 should be presented in methods 1st
line 228- unclear what a second class emergency responder is
line 238- remove personal statement
line 240- unclear what this statement is referring to
line 242- unclear why referring to paramedics when study used firefighters
line 249- possibly tip with 25 in sentence
line 260-261
Conclusion
line 263- mention purpose was to study new firefighters but this was not mentioned in initial purpose statement or title
introduce thoughts on training mannequins - however this was not aim of study
Author Response

(The authors gave the same response as above.)

Round 2
Reviewer 1 Report
Abstract needs to include age and demographic information as well some statistics in the results section of the abstract
Author Response
Comments and Suggestions for Authors
Abstract needs to include age and demographic information as well some statistics in the results section of the abstract
→ An explanation of the subject's characteristics was added to the abstract. Checked for correct English grammar. Thank you for your valuable comments.
Reviewer 2 Report
While some comments from the 1st review were incorporated and/or improved there is still considerable edits to complete.
For the 2nd review I only see that changes to the Intro and the addition of 1-2 sentences in methods and conclusion were added. I believe the initial comments and edits were applicable and warranted for clarity and supported a publishable manuscript.
The current paper continues to be limited in methodology details and clarity, the Introduction is repetitive and confusing. There are grammar and incomplete sentences that add to the confusion. The purpose remains unclear for the reader.
I believe continued revisions and edits are needed and should be completed however this recent revision read rushed and disorganized.
Author Response
Comments and Suggestions for Authors
While some comments from the 1st review were incorporated and/or improved there is still considerable edits to complete.
For the 2nd review I only see that changes to the Intro and the addition of 1-2 sentences in methods and conclusion were added. I believe the initial comments and edits were applicable and warranted for clarity and supported a publishable manuscript.
The current paper continues to be limited in methodology details and clarity, the Introduction is repetitive and confusing. There are grammar and incomplete sentences that add to the confusion. The purpose remains unclear for the reader.
I believe continued revisions and edits are needed and should be completed however this recent revision read rushed and disorganized.
→ Theoretical considerations in the introduction have been added, and repetitive content has been deleted. The introduction and conclusion have been extensively revised. Also, English grammar was checked. Thank you for your valuable comments.